# Gender Differences in the Impact of New-Onset Atrial Fibrillation on Long-Term Risk of Ischemic Stroke after Acute Myocardial Infarction

**DOI:** 10.3390/jcm10215141

**Published:** 2021-11-01

**Authors:** Jeong-Eun Yi, Suk-Min Seo, Sungmin Lim, Eun-Ho Choo, Ik-Jun Choi, Kwan-Yong Lee, Byung-Hee Hwang, Chan-Joon Kim, Mahn-Won Park, Dong-Bin Kim, Sung-Ho Her, Jong-Min Lee, Chul-Soo Park, Pum-Joon Kim, Hee-Yeol Kim, Ki-Dong Yoo, Doo-Soo Jeon, Wook-Sung Chung, Myung-Ho Jeong, Youngkeun Ahn, Kiyuk Chang

**Affiliations:** 1Division of Cardiology, Department of Internal Medicine, Eunpyeong St. Mary’s Hospital, College of Medicine, The Catholic University of Korea, Seoul 06591, Korea; jung30134@naver.com (J.-E.Y.); cardiokim@gmail.com (P.-J.K.); 2Division of Cardiology, Department of Internal Medicine, Uijeongbu St. Mary’s Hospital, College of Medicine, The Catholic University of Korea, Seoul 06591, Korea; mdsungminlim@gmail.com (S.L.); godandsci@catholic.ac.kr (C.-J.K.); leejongm@catholic.ac.kr (J.-M.L.); 3Division of Cardiology, Department of Internal Medicine, Seoul St. Mary’s Hospital, College of Medicine, The Catholic University of Korea, Seoul 06591, Korea; cmcchu@catholic.ac.kr (E.-H.C.); kyle210@naver.com (K.-Y.L.); hbhmac@catholic.ac.kr (B.-H.H.); chungws@catholic.ac.kr (W.-S.C.); kiyuk@catholic.ac.kr (K.C.); 4Division of Cardiology, Department of Internal Medicine, Incheon St. Mary’s Hospital, College of Medicine, The Catholic University of Korea, Seoul 06591, Korea; mrfasthand@catholic.ac.kr (I.-J.C.); coronary@catholic.ac.kr (D.-S.J.); 5Division of Cardiology, Department of Internal Medicine, Daejeon St. Mary’s Hospital, College of Medicine, The Catholic University of Korea, Seoul 06591, Korea; pmw6193@catholic.ac.kr; 6Division of Cardiology, Department of Internal Medicine, Bucheon St. Mary’s Hospital, College of Medicine, The Catholic University of Korea, Seoul 06591, Korea; dbkimmd@catholic.ac.kr (D.-B.K.); cumckhy@catholic.ac.kr (H.-Y.K.); 7Division of Cardiology, Department of Internal Medicine, St. Vincent’s Hospital, College of Medicine, The Catholic University of Korea, Seoul 06591, Korea; hhhsungho@catholic.ac.kr (S.-H.H.); yookd@catholic.ac.kr (K.-D.Y.); 8Division of Cardiology, Department of Internal Medicine, Yeouido St. Mary’s Hospital, College of Medicine, The Catholic University of Korea, Seoul 06591, Korea; charlie@catholic.ac.kr; 9Division of Cardiology, Department of Internal Medicine, Chonnam National University Hospital, Gwangju 61748, Korea; myungho@chollian.net (M.-H.J.); cecilyk@hanmail.net (Y.A.)

**Keywords:** new onset atrial fibrillation, acute myocardial infarction, ischemic stroke, gender differences

## Abstract

Background: Atrial fibrillation (AF) has been identified as a major risk factor for mortality after acute coronary syndrome (ACS). However, the long-term risk of ischemic stroke associated with new-onset atrial fibrillation (NOAF) in ACS remains controversial, and its gender-specific association is unknown. Methods: We analyzed the data of 10,137 ACS survivors included in a multicenter, prospective registry for Korean patients with acute myocardial infarction (AMI) between January 2004 and August 2014. Subjects were categorized into three groups (non-AF vs. NOAF vs. previous AF) based on medical history and electrocardiographic evidence of AF, either at admission or during hospitalization. Results: Among the total study population (72.3% men), 370 patients (3.6%) had NOAF and 130 (1.3%) had previous AF. During a median follow-up of 61 months (interquartile range, 38.8 to 89.3 months), 245 (2.4%) patients (218 (2.3%) non-AF vs. 15 (4.1%) NOAF vs. 12 (9.2%) previous AF, *p* < 0.001) experienced ischemic stroke. After adjustment for confounding variables, both NOAF (adjusted hazard ratio (HR) 1.87, 95% confidence interval (CI) 1.09–3.24, *p* = 0.024) and previous AF (adjusted HR 4.00, 95% CI 2.03–7.87, *p* < 0.001), along with older age, diabetes, current smoker, and previous stroke were independent risk factors of ischemic stroke. In the gender-stratified analysis, men with previous AF but not NOAF had a significantly higher risk of ischemic stroke (adjusted HR 4.14, 95% CI 1.79–9.55, *p* = 0.001) than those without AF. In women, NOAF (adjusted HR 2.54, 95% CI 1.21–5.35, *p* = 0.014) as well as previous AF (adjusted HR 3.72, 95% CI 1.16–11.96, *p* = 0.028) was a strong predictor of ischemic stroke, and the predictive value was comparable to that of previous AF among patients with a CHA_2_DS_2_-VASc score ≥ 2. Conclusions: Both NOAF and previous AF were associated with ischemic stroke after AMI, but the impact of NOAF as a risk factor of ischemic stroke was significant only in women.

## 1. Introduction

Atrial fibrillation (AF) is a common pre-existing comorbidity in patients with acute coronary syndrome (ACS) and is a frequent complication of ACS, with an incidence ranging from 6% to 21% [1,2]. However, new-onset AF (NOAF) in the setting of ACS has shown different clinical characteristics or prognoses from those of previous AF [3], and its underlying mechanism remains unclear [4]. Furthermore, unlike the previous AF that has been relatively well-established as a risk factor for ischemic stroke [5], the association of NOAF complicating ACS with risk of ischemic stroke is controversial [6,7]. In a recently published meta-analysis, Luo J. et al. identified a significantly higher risk of ischemic stroke after ACS in patients with NOAF compared with those in sinus rhythm, but their results were obtained mostly from observational studies that examined in-hospital or short-term outcomes with different definitions of NOAF [8].

Although AF is an independent predictor of ischemic stroke in men and women, women are at greater risk for stroke than men despite being treated with anticoagulants [9,10,11]. It has been thought that the higher risk of stroke in women with AF is attributed to older age, worse clinical profiles and under-treatment for co-morbidities compared to men [10,11,12,13]. Similarly, gender differences in in-hospital management and outcomes following ACS have been observed, with less invasive therapy and a higher in-hospital mortality rate in women than in men [14,15]. However, the exact pathophysiology for gender differences in the effect of AF on the risk of stroke is still unknown [16], and previous studies investigating gender-specific association of AF with the risk of stroke have been conducted in the general population [9,10,11,12,13]. Moreover, the differential impact of NOAF after ACS on the risk of ischemic stroke in men and women has not been determined. Thus, we sought to estimate the long-term risk of ischemic stroke associated with NOAF complicating ACS and compare it with that of previous AF and further evaluated whether its prognostic burden differs between men and women using data from a multicenter, prospective registry for Korean patients with acute myocardial infarction (AMI).

## 2. Methods

### 2.1. COREA-AMI Registry

We conducted a prospective cohort study using data from the CardiOvascular Risk and idEntification of potential high-risk population in AMI (COREA-AMI) registry. The details of the registry have been published [17]. In brief, the COREA-AMI is a prospective, multicenter registry used to evaluate the outcomes of AMI patients in real-world clinical practice and involved consecutive patients (age > 20 years) who presented to nine nation-wide hospitals with a final diagnosis of AMI and were treated with percutaneous coronary intervention (PCI) using stents from January 2004 to August 2014. The diagnostic criteria for AMI were based on the universal definition of myocardial infarction [18], and PCI was performed according to the standard guidelines. Clinical and outcome data were collected by individual investigators using the protocol definitions, and angiographic and procedural data were obtained by an independent interventional cardiologist. Laboratory data were obtained at admission and echocardiographic data were obtained within 24 h of admission. Each enrolled patient was followed up regularly via outpatient clinical visit or telephone call. All adverse clinical events were confirmed by source documents such as medical records or telephone interviews and were adjudicated centrally by a committee of the Cardiovascular Center of Seoul St. Mary’s Hospital, Seoul, Korea. Validation of complete follow-up data, including information on censored survival data and causes of death, were performed using data categorized by unique identification numbers from the Korea National Statistics Office of Statistics. The study protocol adhered to the ethical principles of the Declaration of Helsinki for medical research involving human subjects, and written informed consent was obtained from all participants before enrollment. This registry was reviewed and approved by the institutional review board of each participating hospital and has been registered on ClinicalTrials.gov (https://clinicaltrials.gov/ (accessed on 11 March 2015)) (study ID: NCT02385682). 

### 2.2. Study Population and Variables

Of the total of 10,719 patients enrolled in this registry, those with a missing electrocardiographic (ECG) diagnosis for AF or history of AF not documented in the medical record (*n* = 30) and those with in-hospital death (*n* = 552) were excluded. The remaining 10,137 patients were included in our study and were categorized into two groups according to sex (7327 (72.3%) men and 2810 (27.7%) women) (Figure 1). AF was defined as an irregular heart rhythm with the absence of discrete atrial activation on 12-lead ECG [5]. Patients with AF were classified as either NOAF or previous AF based on medical history and presence of AF newly documented on in-hospital ECG. Patients with AF developing for the first time after ACS were considered to have NOAF, and those with AF diagnosed prior to ACS admission were categorized as having previous AF. These two groups were compared to the non-AF group, who did not show AF either at admission or during hospitalization and had no history of AF. The primary endpoint of this study was ischemic stroke occurring after hospital discharge. Ischemic stroke was defined as an acute neurological deficit of ischemic vascular origin lasting more than 24 h or leading to death, which was confirmed by a neurologist or brain imaging modality [19]. The stroke risk stratification was assessed using the CHA_2_DS_2_-VASc score (congestive heart failure [HF], hypertension, age ≥ 75 years (doubled), type 2 diabetes, previous stroke, or transient ischemic attack (doubled), vascular disease, age 65–74 years, and sex), which was calculated for each patient based on demographic and clinical information at enrollment, and patients with ≥2 points were classified as a high-risk group for stroke in our study [5].

### 2.3. Statistical Analysis

Continuous variables are expressed as mean ± standard deviation (SD) or median (interquartile range, [IQR]) and are compared by independent sample *t*-test or Mann–Whitney *U*-test, where appropriate. Categorical variables are expressed as percentage and compared by a Chi-square or Fisher’s exact test, as appropriate. The cumulative incidence rates of ischemic stroke among the 3 subgroups (non-AF vs. NOAF vs. previous AF) were estimated using Kaplan–Meier curves and were compared by log-rank test. Cox proportional hazards models with the backward elimination selection method were used to investigate the association between each AF subgroup and the risk of ischemic stroke following ACS, and the hazard ratios (HRs) along with their 95% confidence intervals (CIs) for each model were calculated. Multivariate Cox regression analysis was performed after adjusting for covariates of age; sex; body mass index (BMI); diabetes; hypertension; dyslipidemia; chronic kidney disease; current smoker; previous history of HF, MI, or stroke; Killip class ≥ 2 at admission; ST-segment elevation myocardial infarction (STEMI); extent of coronary artery disease (CAD); and dual antiplatelet therapy (DAPT) and/or oral anticoagulant (OAC) at discharge. To evaluate gender differences in the risk of stroke according to AF subgroup, Cox regression analyses were performed separately in men and women using the clinical variables mentioned above; in women, we additionally adjusted for menopause. All statistical analyses were performed using SAS version 9.4 (SAS Institute Inc., Cary, NC, USA) and two-sided *p* values < 0.05 were considered statistically significant.

## 3. Results

### 3.1. Patient Characteristics

Our study included 10,137 patients, of which 7327 (72.3%) were men and 2810 (27.7%) were women. The baseline characteristics of the study population by gender are presented in Table 1. The mean age was 63.2 ± 12.7 years, and women were older than men (71.0 ± 10.4 vs. 60.2 ± 12.2 years, *p* < 0.001). Compared with men, women were more likely to have diabetes, hypertension, and previous history of HF and stroke; however, men had a higher BMI and a higher prevalence of current smoker, family history of CAD, and previous history of MI than women. Women had a lower diastolic blood pressure (BP) level, higher heart rate, lower left ventricular ejection fraction (LVEF), and higher levels of HbA1c and low-density lipoprotein (LDL) cholesterol at admission than men and were more likely to present with a higher Killip class (≥2). Although men more frequently presented with STEMI and higher peak levels of creatine kinase, creatine kinase-myocardial band and troponin I than women, extensive CAD was found more commonly in women. Among the overall study population, 370 patients (3.6%) had NOAF and 130 (1.3%) had previously known AF. The prevalence of AF (168 (6.0%) vs. 332 (4.5%), *p* = 0.003), including NOAF (118 (4.2%) vs. 252 (3.4%), *p* = 0.068) and previous AF (50 (1.8%) vs. 80 (1.1%), *p* = 0.006), was significantly higher in women than in men, and women exhibited a higher mean CHA_2_DS_2_-VASc score (4.2 ± 1.8 vs. 1.8 ± 1.7, *p* < 0.001). At discharge, women were less likely to be treated with statins, beta-blockers, and angiotensin-converting enzyme inhibitors or angiotensin receptor blockers than men; however, OACs were prescribed more frequently in women than in men, especially warfarin (84 (3.0%) vs. 159 (2.2%), *p* = 0.016) and the combination of OAC with DAPT (82 (2.9%) vs. 154 (2.0%), *p* = 0.013). There was no significant difference in the prescription rate of aspirin, P_2_Y_12_ inhibitors, and DAPT between men and women. 

### 3.2. Gender Differences in the Incidence of Ischemic Stroke

During a median follow-up of 61 months (interquartile range, 38.8 to 89.3 months), a total of 245 (2.4%) patients experienced ischemic stroke after hospital discharge, and its incidence was higher in women than in men (83 (3.0%) vs. 162 (2.2%), *p* = 0.029). Overall, ischemic stroke most commonly occurred in patients with previous AF, followed by patients with NOAF, and those without AF (12 (9.2%) vs. 15 (4.1%) vs. 218 (2.3%), *p* < 0.001). Similar results were observed in both genders (all *p* < 0.05), but in men, the incidence of ischemic stroke did not differ significantly between patients with NOAF and those without AF (*p* = 0.486) (Figure 2). The Kaplan–Meier curves for ischemic stroke illustrate the significant differences in cumulative incidence among the three groups (Figure 3). In the overall population, patients with NOAF exhibited an intermediate trend of cumulative incidence rate for ischemic stroke (log-rank, *p* < 0.001) (Figure 3A). In men, patients with NOAF revealed a similar clinical outcome as those without AF (log-rank, *p* = 0.354) and showed a lower cumulative incidence rate for ischemic stroke than patients with previous AF (log-rank, *p* = 0.027) (Figure 3B). In women, patients with NOAF had a higher cumulative incidence rate for ischemic stroke than those without AF (log-rank, *p* = 0.003) and demonstrated no significant difference in cumulative incidence rate compared to patients with previous AF (log-rank, *p* = 0.133) (Figure 3C).

### 3.3. Impact of NOAF on the RIsk of Ischemic Stroke by Gender

Multivariate Cox regression analyses for the association of NOAF with the risk of ischemic stroke are presented in Table 2. The results of unadjusted and adjusted analyses are provided in Appendix A. After adjustment for confounding factors, both NOAF (adjusted HR 1.87, 95% CI 1.09–3.24, *p* = 0.024) and previous AF (adjusted HR 4.00, 95% CI 2.03–7.87, *p* < 0.001), old age, diabetes, current smoker, and previous stroke were associated with ischemic stroke. When analyzing the predictors of ischemic stroke in men, previous AF (adjusted HR 4.14, 95% CI 1.79–9.55, *p* = 0.001), along with old age, lower BMI, diabetes, current smoker, and previous stroke were independent predictors of ischemic stroke, but NOAF was not (*p* = 0.439). Meanwhile, in women, NOAF (adjusted HR 2.54, 95% CI 1.21–5.35, *p* = 0.014) was a significant predictor of ischemic stroke, as was previous AF (adjusted HR 3.72, 95% CI 1.16–11.96, *p* = 0.028), old age, and Killip class ≥ 2. Subgroup analyses for patients with CHA_2_DS_2_-VASc score ≥ 2 were performed as shown in Table 3. The AF itself significantly increased the risk of ischemic stroke (2.07- and 4.50-fold risk increases in NOAF and previous AF, respectively) and the risk tended to be higher in previous AF than in NOAF (*p* = 0.074). In men, previous AF showed a higher risk of ischemic stroke than did NOAF (adjusted HR 5.07, 95% CI 2.20–11.70, *p* < 0.001) or absence of AF (adjusted HR 4.79, 95% CI 1.93–11.88, *p* = 0.001), and NOAF did not increase the risk of ischemic stroke (*p* = 0.441). However, in women, both NOAF (adjusted HR 2.57, 95% CI 1.22–5.41, *p* = 0.013) and previous AF (adjusted HR 3.80, 95% CI 1.18–12.23, *p* = 0.025) significantly increased the risk of ischemic stroke compared with the absence of AF, and the risk did not differ between the two groups (*p* = 0.590).

## 4. Discussion

The major findings of this study among patients with AMI undergoing PCI were as follows. First, the incidence of ischemic stroke after AMI was significantly higher in patients with previous AF than in those with NOAF or without AF. The incidence rate was not significantly different between male patients with NOAF and without AF, but female patients with NOAF showed an intermediate incidence rate for ischemic stroke between patients with previous AF and those without AF. Second, both NOAF and previous AF were associated with increased risk of ischemic stroke. However, NOAF was a strong independent risk factor of ischemic stroke in women but not in men, and its predictive value for ischemic stroke was comparable to that of previous AF.

In the present study, we investigated long-term risk of ischemic stroke associated with post-MI NOAF, and this risk was compared with that of previous AF. Although numerous studies have reported a significant association between increased mortality and post-MI NOAF, clinical data on the risk of ischemic stroke during long-term follow-up are limited, and their results are conflicting [6,7,20,21]. In our study, post-MI NOAF was an independent predictor of ischemic stroke, along with previous AF. Patients with NOAF were older and tended to have more comorbidities with a significantly higher CHA_2_DS_2_-VASc score, lower LVEF, and greater left atrial dimension than those without AF (Appendix A). The pathophysiology of NOAF in the setting of ACS is likely multifactorial, and it is unclear whether post-MI NOAF itself is a cardio-embolic risk factor for stroke or just an epiphenomenon of underlying stroke-related comorbid conditions [4]. However, we assessed the risk of subsequent ischemic stroke after hospital discharge for AMI, and this risk remained significant even after adjusting for multiple risk factors of stroke. Consistent with previous studies that have shown an increased risk of ischemic stroke in patients with NOAF complicating ACS [8], our results revealed a significant association between ischemic stroke and post-MI NOAF. It is well known that previous AF reflects the greater burden of cardiovascular diseases and risk factors and induces adverse cardiac remodeling due to the longer duration of the arrhythmia than in NOAF [22]. In this study, we also found baseline characteristics of patients with previous AF that were different from those with NOAF (Appendix A), and the risk of ischemic stroke seemed to be higher in patients with previous AF than in those with NOAF. Given the etiology of ischemic stroke, including atherothrombosis and cardiac embolism [23], patients with previous AF might have a higher probability of experiencing both types of ischemic stroke compared to those with NOAF.

We performed subgroup analyses in men and women to examine the gender differences in the effect of post-MI NOAF on the long-term risk of ischemic stroke. Although the prevalence of AF is higher in men than in women [24], women were more likely to experience stroke compared with men [9,10]. However, the underlying mechanism for this gender difference in stroke risk among patients with AF remains unclear. In several large cohort studies that investigated AF-related stroke risk by gender, women usually showed worse conventional risk profiles for stroke compared to men, which has been postulated as a potential cause for the increased risk of stroke in women [16]. Some authors reported that gender-related differences in cardiovascular remodeling can cause alterations in arterial hemodynamics and wall shear stress, leading to endothelial dysfunction that has been thought to play an important role in the pathogenesis of ischemic stroke [25]. The hypercoagulable state linked to the hormonal differences and increased systemic inflammatory or pro-thrombotic biomarkers, especially after menopause, has been suggested to contribute to the differential effect of AF on stroke in men and women [26]. In our study of patients presenting with AMI, female gender itself was not an independent predictor of stroke, but post-MI NOAF was a strong risk factor for ischemic stroke only in women. We could not determine what could cause this observed difference in stroke risk associated with post-MI NOAF between men and women. In our study, women with NOAF had more risk factors for stroke and exhibited a significantly higher CHA_2_DS_2_-VASc score than men with NOAF. Interestingly, women with NOAF tended to have similar comorbid profiles as those with previous AF, whereas men with NOAF showed more favorable clinical features than those with previous AF (Appendix A). These findings suggest that, among patients with AMI, a relatively high comorbidity burden in women with NOAF is associated with increased risk of ischemic stroke during long-term follow-up after AMI.

Several studies have suggested that women are less likely to be treated with OACs for AF relative to men. In a prospective cohort study of Canadian patients with AF, Humphries KH et al. found that elderly women were about 50% less likely to receive warfarin than elderly men [27]. A recently published study on patients with AF at high risk of stroke showed a substantially lower prescription rate of OACs, including new oral anticoagulants (NOACs) in women than in men and reported that this gender difference of AF management can contribute to a higher risk of ischemic stroke in women compared to men [13]. On the other hand, some cohort studies of anticoagulated AF patients have reported consistently a greater risk of ischemic stroke in women than in men, despite no difference in the use of OACs [28,29]. In our study, even though women with NOAF were more likely to receive warfarin at discharge than were men with NOAF (22.9% vs. 13.1%, *p* = 0.018), they tended to experience more ischemic stroke events during follow-up (Figure 2), and a significant association between ischemic stroke and post-MI NOAF was found in women but not in men. Previously, Sullivan R et al. found that women had a greater risk of ischemic stroke than men among patients treated with warfarin for AF, which was associated with a lower time in therapeutic range (TTR) for warfarin treatment, representing poor anticoagulation control in women [30]. Furthermore, a meta-analysis by Pancholy SB et al. revealed a net clinical benefit of NOACs over warfarin in women with AF, indicating the importance of a constant anticoagulant effect for stroke prevention among women with AF [31]. Although the TTR level was not included in our data, it is speculated that gender-specific differences in pharmacodynamic properties and therapeutic efficacy of warfarin could be one explanation for our results. Currently, patients with ACS and AF at high risk of thromboembolism (CHA_2_DS_2_-VASc score ≥ 2) are recommended strongly to be treated with OACs [5]; however, for patients with NOAF undergoing PCI for ACS, use of OACs in addition to DAPT agents is still suboptimal. In a subgroup analysis of our patients with CHA_2_DS_2_-VASc score ≥ 2, we confirmed that post-MI NOAF significantly increased the risk of ischemic stroke, similar to previous AF in women. Our results suggest that women with post-MI NOAF should be managed more aggressively according to the guidelines, targeting not only effective anticoagulation, but also control of modifiable risk factors for stroke following PCI for AMI. This study is meaningful in that long-term risk of ischemic stroke associated with post-MI NOAF was explored, particularly focusing on the gender difference in patients with AMI.

The present study has several limitations. First, this is an observational cohort study in which there might be potential confounding factors that could influence the clinical outcomes. Moreover, our results were derived from AMI-specific registry data, and detailed information regarding AF, such as type (paroxysmal, persistent, or permanent) and duration, and antiarrhythmic therapy, were not assessed. However, this is a large prospective, multicenter registry reflecting a real-world routine clinical procedure, and all participants, regardless of gender, underwent PCI following standard recommendations. Second, although we excluded patients with a history of AF based on medical records, there was a possibility that NOAF was undiagnosed asymptomatic AF prior to hospitalization rather than, NOAF occurring in the setting of AMI. In addition, continuous ECG monitoring for detection of silent AF that has been reported up to 16% after AMI was not performed [32]. Nevertheless, we could identify the unique clinical and echocardiographic features of patients with NOAF that are clearly differentiated from those of patients with previous AF and without AF. Third, our data were collected between 2004 and 2014, and might not reflect updated practical guidelines for AF, including risk assessment of stroke using the CHA_2_DS_2_-VASc score or treatment with NOACs. This could explain the lower prescription rate of NOACs compared with warfarin in our study. Finally, there was lack of data about gender differences in adherence to treatment for AF, especially OACs during follow-up, which can affect the long-term outcome.

## 5. Conclusions

In this study of patients with AMI undergoing PCI, post-MI NOAF was significantly associated with increased risk of ischemic stroke after hospital discharge, especially among women, and the risk did not differ from that of previous AF in a subgroup at high risk for thromboembolism. These results indicate gender differences in the effect of NOAF complicating AMI on the long-term risk of ischemic stroke, and further suggest that a gender-specific risk assessment and optimal preventive strategies for ischemic stroke associated with post-MI NOAF might be needed.

## Figures and Tables

**Figure 1 jcm-10-05141-f001:**
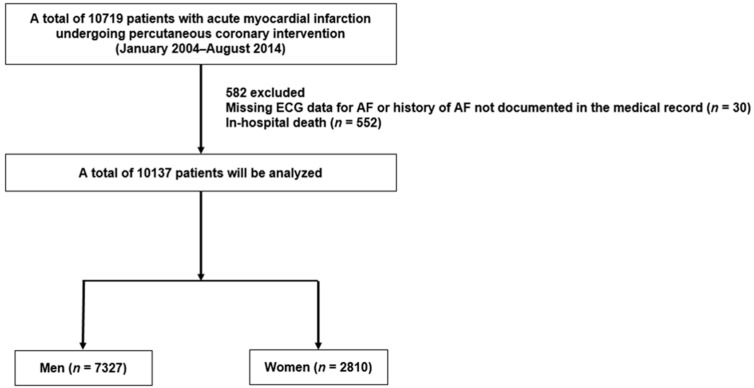
Study flow chart.

**Figure 2 jcm-10-05141-f002:**
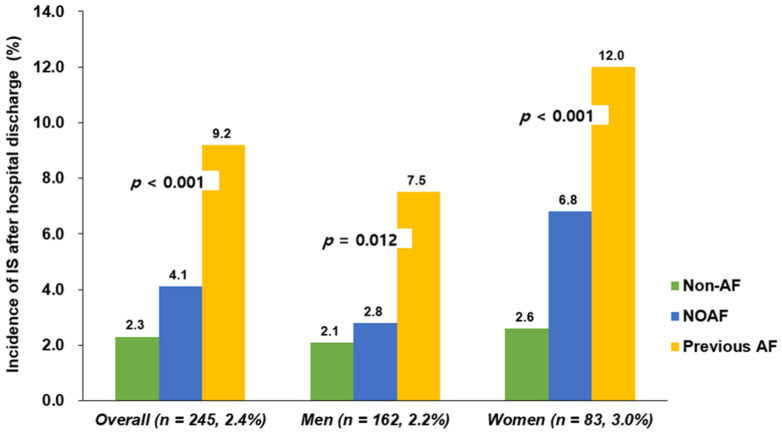
Incidence of ischemic stroke (IS) during follow-up after acute myocardial infarction according to onset time of atrial fibrillation (AF) among the overall population, men, and women.

**Figure 3 jcm-10-05141-f003:**
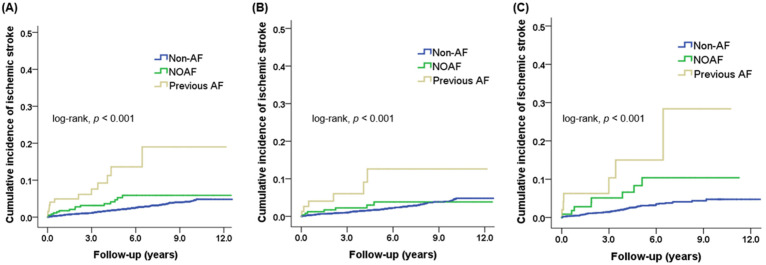
Kaplan–Meier curves for the cumulative incidence of ischemic stroke after acute myocardial infarction according to onset time of atrial fibrillation in (**A**) the overall population, (**B**) men, and (**C**) women.

**Table 1 jcm-10-05141-t001:** Baseline characteristics.

Variables	Total (*n* = 10,137)	Men (*n* = 7327)	Women (*n* = 2810)	*p* Value
Age (years)	63.2 ± 12.7	60.2 ± 12.2	71.0 ± 10.4	<0.001
Body mass index (kg/m^2^)	24.2 ± 3.3	24.4 ± 3.1	23.7 ± 3.6	<0.001
Diabetes mellitus (%)	3168 (31.3)	2079 (28.4)	1089 (38.8)	<0.001
Hypertension (%)	5250 (51.8)	3367 (46.0)	1883 (67.0)	<0.001
Dyslipidemia (%)	1635 (16.1)	1156 (15.8)	479 (17.0)	0.120
Chronic kidney disease (%)	198 (2.0)	132 (1.8)	66 (2.3)	0.075
Current smoker (%)	4136 (40.8)	3864 (52.7)	272 (9.7)	<0.001
Family history of CAD (%)	300 (3.0)	251 (3.4)	49 (1.7)	<0.001
Previous HF (%)	126 (1.2)	65 (0.9)	61 (2.2)	<0.001
Previous MI (%)	416 (4.1)	319 (4.4)	97 (3.5)	0.040
Previous PAD (%)	55 (0.5)	43 (0.6)	12 (0.4)	0.327
Previous stroke (%)	707 (7.0)	464 (6.3)	243 (8.6)	<0.001
Menopause, *n* (%)	2001 (19.7)	-	2001 (71.2)	NA
SBP at admission (mmHg)	129 ± 26	129 ± 26	129 ± 27	0.671
DBP at admission (mmHg)	79 ± 16	79 ± 16	77 ± 16	<0.001
HR at admission (mmHg)	79 ± 19	78 ± 18	80 ± 20	<0.001
Killip class ≥ 2 (%)	2351 (23.2)	1531 (20.9)	820 (29.2)	<0.001
LV ejection fraction (%)	53.4 ± 11.1	53.7 ± 10.9	52.6 ± 11.5	<0.001
LA AP diameter (mm)	36.9 ± 6.2	36.9 ± 6.1	37.0 ± 6.5	0.917
eGFR by MDRD (mL/min/1.73 m^2^)	64.2 ± 23.9	62.2 ± 22.0	69.3 ± 27.6	<0.001
HbA1c (%)	6.6 ± 1.6	6.6 ± 1.6	6.8 ± 1.5	<0.001
Total cholesterol (mg/dL)	178.1 ± 43.4	176.8 ± 41.9	181.4 ± 47.0	<0.001
Triglyceride (mg/dL)	124.7 ± 92.4	128.4 ± 99.3	114.9 ± 70.3	<0.001
HDL cholesterol (mg/dL)	40.9 ± 10.9	40.1 ± 10.5	43.1 ± 11.8	<0.001
LDL cholesterol (mg/dL)	113.6 ± 37.9	112.9 ± 36.8	115.3 ± 40.6	0.011
Peak CK (U/L)	443 (149,1435)	491 (160,1612)	330 (120,1032)	<0.001
Peak CK-MB (µg/mL)	47.6 (14.0,145.4)	54.8 (15.3,156.2)	34.0 (11.2,113.1)	<0.001
Peak TnI (ng/mL)	22.6 (4.5,52.0)	25.0 (5.0,55.1)	14.7 (3.5,50.0)	<0.001
STEMI (%)	5430 (53.6)	4091 (55.8)	1339 (47.7)	<0.001
Extent of CAD (%)				
One vessel disease	4658 (46.0)	3490 (47.6)	1168 (41.6)	<0.001
Two vessel disease	3310 (32.7)	2382 (32.5)	928 (33.0)	0.621
Three vessel disease	2169 (21.4)	1455 (19.9)	714 (25.4)	<0.001
CABG (%)	20 (0.2)	14 (0.2)	6 (0.2)	0.818
Total AF (%)	500 (4.9)	332 (4.5)	168 (6.0)	0.003
New onset AF	370 (3.6)	252 (3.4)	118 (4.2)	0.068
Previous AF	130 (1.3)	80 (1.1)	50 (1.8)	0.006
CHA_2_DS_2_-VASc score	2.5 ± 2.0	1.8 ± 1.7	4.2 ± 1.8	<0.001
Medications at discharge				
Aspirin (%)	9964 (98.3)	7209 (98.4)	2755 (98.0)	0.228
P_2_Y_12_ inhibitor (%)	9690 (95.6)	7004 (95.6)	2686 (95.6)	0.893
DAPT (%)	9620 (94.9)	6967 (95.1)	2653 (94.4)	0.254
Anticoagulant (%)	257 (2.5)	167 (2.2)	90 (3.2)	0.008
Warfarin	243 (2.4)	159 (2.2)	84 (3.0)	0.016
NOAC	14 (0.1)	8 (0.1)	6 (0.2)	0.233
DAPT plus anticoagulant (%)	236 (2.3)	154 (2.0)	82 (2.9)	0.013
Statin (%)	9741 (96.1)	7070 (96.5)	2671 (95.1)	<0.001
Beta-blocker (%)	8819 (87.0)	6411 (87.5)	2408 (85.7)	0.026
ACE inhibitor or ARB (%)	9265 (91.4)	6733 (91.9)	2532 (90.1)	0.006

The values are mean ± standard deviation (SD) or *n* (%). Abbreviations: CAD, coronary artery disease; HF, heart failure; MI, myocardial infarction; PAD, peripheral arterial disease; NA, non-applicable; SBP, systolic blood pressure; DBP, diastolic blood pressure; HR, heart rate; LV, left ventricular; LA, left atrial; AP, anterior-posterior; eGFR by MDRD, estimated glomerular filtrate rate using modification of diet in renal disease; HDL, high density lipoprotein; LDL, low density lipoprotein; CK, creatine kinase; CK-MB, creatine kinase-myocardial band; TnI, troponin I; AF, atrial fibrillation; STEMI, ST-segment-elevation myocardial infarction; CABG, coronary artery bypass grafting; DAPT, dual antiplatelet; NOAC, new oral anticoagulant; ACE, angiotensin-converting enzyme inhibitor; ARB, angiotensin receptor blocker.

**Table 2 jcm-10-05141-t002:** Cox-regression analyses of the associations between each AF subgroup and the risk of ischemic stroke after hospital discharge.

Variables	Overall		Men		Women	
	Adjusted HR (95% CI)	*p* Value	Adjusted HR (95% CI)	*p* Value	Adjusted HR (95% CI)	*p* Value
Non-AF	1 (reference)	NA	1 (reference)	NA	1 (reference)	NA
NOAF	1.87 (1.09–3.24)	0.024	1.39 (0.61–3.17)	0.439	2.54 (1.21–5.35)	0.014
Previous AF	4.00 (2.03–7.87)	<0.001	4.14 (1.79–9.55)	0.001	3.72 (1.16–11.96)	0.028

Abbreviations: HR, hazard ratio; CI, confidence interval; AF, atrial fibrillation; NOAF, new onset atrial fibrillation; NA, non-applicable.

**Table 3 jcm-10-05141-t003:** Risk for ischemic stroke in patients with CHA_2_DS_2_-VASc score ≥ 2 according to AF subgroups.

Subjects	Non-AF	NOAF	NOAF vs. Non-AF (Ref.) Adjusted HR (95% CI); *p* Value	Previous AF	Previous AF vs. Non-AF (Ref.) Adjusted HR (95% CI); *p* Value	Previous AF vs. NOAF (Ref.) Adjusted HR (95% CI); *p* Value
Overall (*n* = 6130)	2.8%	5.0%	2.07 (1.17–3.66); *p* < 0.001	9.7%	4.50 (2.28–8.85); *p* < 0.001	2.17 (0.93–5.09); *p* = 0.074
Men (*n* = 3506)	2.9%	3.6%	1.45 (0.56–3.74); *p* = 0.441	9.2%	4.79 (1.93–11.88); *p* = 0.001	5.07 (2.20–11.70); *p* < 0.001
Women (*n* = 2624)	2.7%	7.0%	2.57 (1.22–5.41); *p* = 0.013	10.4%	3.80 (1.18–12.23); *p* = 0.025	1.45 (0.38–5.57); *p* = 0.590

Abbreviations: AF, atrial fibrillation; NOAF, new onset atrial fibrillation; HR, hazard ratio; CI, confidence interval.

## Data Availability

Data related to the study are available upon request to the corresponding author.

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
