# Peer review of "Gender Differences in the Impact of New-Onset Atrial Fibrillation on Long-Term Risk of Ischemic Stroke after Acute Myocardial Infarction"

_jcm, 2021, doi:10.3390/jcm10215141_

Round 1

Reviewer 1 Report

The authors investigated whether its prognostic burden differs between men and women using data from a multicenter, prospective registry for Korean patients with AMI. The major findings of this study were as follows. First, the incidence of ischemic stroke after AMI was significantly higher in patients with previous AF than in those with NOAF or without AF. The incidence rate was not significantly different between male patients with NOAF and without AF, but female patients with NOAF showed an intermediate incidence rate for ischemic stroke between patients with previous AF and those without AF. Second, both NOAF and previous AF were associated with increased risk of ischemic stroke. However, NOAF was a strong independent risk factor of ischemic stroke in women but not in men, and its predictive value for ischemic stroke was comparable to that of previous AF.

This is a multicenter study of post-discharge ischemic stroke after AMI in patients with NOAF, Non-AF, and Previous AF, including gender differences. The number of cases and the detailed patient backgrounds and results are very impressive. Most importantly, this prospective study represents a great deal of effort; however, the impact is lost by some problems as indicated below. This manuscript has some points that need to be clarified for publication in this journal.

  1. All prospective studies are worthy of praise, but unfortunately, there have been reported investigating on the correlation between AF and ischemic stroke in AMI patients. This study seems to lack detailed description, evaluation, and consideration of the correlation between gender gap.
  2. The “patients presenting with AMI” part of this title should be changed because this work will describe the long-term outcome of patients with AMI.
  3. To avoid misunderstanding, please add the patient flow chart of the study population in figure 1.

Author Response

Reviewer #1

The authors investigated whether its prognostic burden differs between men and women using data from a multicenter, prospective registry for Korean patients with AMI. The major findings of this study were as follows: First, the incidence of ischemic stroke after AMI was significantly higher in patients with previous AF than in those with NOAF or without AF. The incidence rate was not significantly different between male patients with NOAF and without AF, but female patients with NOAF showed an intermediate incidence rate for ischemic stroke between patients with previous AF and those without AF. Second, both NOAF and previous AF were associated with increased risk of ischemic stroke. However, NOAF was a strong independent risk factor of ischemic stroke in women but not in men, and its predictive value for ischemic stroke was comparable to that of previous AF.

This is a multicenter study of post-discharge ischemic stroke after AMI in patients with NOAF, Non-AF, and Previous AF, including gender differences. The number of cases and the detailed patient backgrounds and results are very impressive. More importantly, this prospective study represents a great deal of effort; however, the impact is lost by some problems as indicated below. This manuscript has some points that need to be clarified for publication in this journal.

1) All prospective studies are worthy of praise, but unfortunately, there have been reported investigating on the correlation between AF and ischemic stroke in AMI patients. This study seems to lack detailed description, evaluation, and consideration of the correlation between gender gap.

Response: That’s a very good point. Although we showed the results from the overall population, the aim of this study was to investigate gender-specific associations between NOAF and long-term risk of ischemic stroke in patients AMI. Actually, through the analysis of the overall population, we wanted to confirm the results of the previously published studies and suggest the need for gender-specific consideration. The authors fully agreed that there was a lack of description, evaluation, and consideration of gender gap in the effect NOAF on risk of ischemic stroke. We additionally described the background for the risk of AF-related ischemic stroke focusing on gender differences and made an effort to emphasize the clinical significance of our study in Introduction section as below;

“Although AF is an independent predictor of ischemic stroke in men and women, women are at greater risk for stroke than men despite being treated with anticoagulants. It has been thought that the higher risk of stroke in women with AF is attributed to older age, worse clinical profiles and under-treatment for co-morbidities compared to men. Similarly, gender differences in in-hospital management and outcomes following ACS have been observed, with less invasive therapy and a higher in-hospital mortality rate in women than in men. However, the exact pathophysiology for gender differences in the effect of AF on the risk of stroke is still unknown, and previous studies investigating gender-specific association of AF with the risk of stroke have been conducted in the general population. Moreover, the differential impact of NOAF after ACS on the risk of ischemic stroke in men and women has not been determined.”

In addition, the subtitles of the Result section have been revised to emphasize the gender differences as below;

“Gender differences in the incidence of ischemic stroke”

“Impact of NOAF on the risk of ischemic stroke by gender”

2.) The “patients presenting with AMI” part of this title should be changed because this work will describe the long-term outcomes of patients with AMI.

Response: Thank you for your advice. As you pointed out, the title has been revised as “Gender differences in the impact of new-onset atrial fibrillation on long-term risk of ischemic stroke after acute myocardial infarction

3) To avoid misunderstanding, please add the patient flow chart of the study population in figure 1.

Response: Thank you for your kind consideration. As you pointed out, we added the study flow chart of the study population in Method section.

Reviewer 2 Report

I’m honored to review the original article entitled “Gender differences in the impact of new-onset atrial fibrillation on the long-term risk of ischemic stroke among patients presenting with acute myocardial infarction” by Jeong-Eun et al.

They investigated the risk of ischemic stroke associated with NOAF complicating ACS and its prognostic impact between men and women using data from COREA-AMI. Based on the results of multivariate analysis, they concluded that both NOAF and previous AF were associated with ischemic stroke after AMI, but the impact of NOAF as a risk factor of ischemic stroke was significant only in women.

This is an interesting work. It is important to examine the significance of NOAF after AMI and whether it differ between men and women.

I have the following comments.

Major comments:

If NOAF is transient due to ACS invasion, the risk of stroke is not higher than nonAF, as in the men in this study, but if it is a manifestation of the original condition due to ACS, the risk is higher than nonAF, as in the women. Based on the patient background, it is likely that the woman was older and had asymptomatic AF beforehand. It is necessary to deny that asymptomatic AF in women is just manifesting itself. Please check the risk stratified by age and sex, for example, men and women under 60, in 60-70, over 70.

How long is hospital stay? Can NOAF be diagnosed even if it has been a long time since the onset of ACS as long as the patient is in the hospital? The longer the time between the onset of ACS and the diagnosis of NOAF, the greater the likelihood that transient AF will be mixed with asymptomatic AF.

Minor comments:

When were the echo data and blood test data obtained? Please describe it in Method section or in Table 1.

Is there a measure of infarct size such as CK?

Why do so few patients with AF take anticoagulation?

Is P<0.0001 P<0.001? Only this part has one more digit.

Author Response

Reviewer #2

I’m honored to review the original article entitled “Gender differences in the impact of new-onset atrial fibrillation on the long-term risk of ischemic stroke among patients presenting with acute myocardial infarction” by Jeong-Eun et al.

They investigated the risk of ischemic stroke associated with NOAF complicating ACS and its prognostic impact between men and women using data from COREA-AMI. Based on the results of multivariate analysis, they concluded that both NOAF and previous AF were associated with ischemic stroke after AMI, but the impact of NOAF as a risk factor of ischemic stroke was significant only in women.

This is an interesting work. It is important to examine the significance of NOAF after AMI and whether it differs between men and women.

I have following comments.

Major comments:

1) If NOAF is transient due to ACS invasion, the risk of stroke is not higher than non-AF, as in the men in this study, but if it is a manifestation of the original condition due to ACS, the risk is higher than non-AF, as in the women. Based on the patient background, it is likely that the woman was older and had asymptomatic AF beforehand. It is necessary to deny that asymptomatic AF in women is just manifesting itself. Please check the risk stratified by age and sex, for example, men and women under 60, in 60-70, over 70.

Response: Thank you for your good advice. As you pointed out, we analyzed the risk of ischemic stroke associated with NOAF in subgroups stratified by age (65 and > 65) and sex (men and women). At first, we tried to analyzed the risk according to age (< 60, 60-70, > 70) as you recommended; however, the number of female patients under 60 years and their stroke event was too small to be analyzed. As a result, we found the significant associations between NOAF and the risk of ischemic stroke in women aged 65, as well as those aged > 65. Previous studies have demonstrated that the higher risk of ischemic stroke in women with AF is associated with a higher clinical risk profile, particularly for old age, compared with men (1-4). In addition, women with ACS were more likely to have cardiovascular risk factors, with worse clinical outcomes than men (5, 6). However, the exact mechanism for differential effect of AF on the risk of stroke still remains unclear and moreover, the underlying pathology for NOAF in the setting of ACS is also controversial. Our findings suggest that there might be differential impact of NOAF on the risk of ischemic stroke between men and women, and the predicting value of NOAF for ischemic stroke is significant only in women. These results are as below:

Subgroups

Univariate analysis

Multivariate analysis

HR (95% CI)

P

HR (95% CI)

P

Men (n=7,327)

Age ≤ 65 (n=4,759)

Non-AF

1 (reference)

1 (reference)

NOAF

0.930 (0.229, 3.783)

0.919

NS

Previous AF

5.424 (1.332, 22.084)

0.018

7.388 (1.767,30.884)

0.006

Age > 65 (n=2,568)

Non-AF

1 (reference)

1 (reference)

NOAF

1.537 (0.620, 3.814)

0.354

NS

Previous AF

4.034 (1.466, 11.099)

0.007

4.363 (1.577, 12.075)

0.005

Women (n=2,810)

Age ≤ 65 (n=742)

Non-AF

1 (reference)

1 (reference)

NOAF

6.875 (1.505, 31.411)

0.013

4.792 (1.005, 22.840)

0.049

Previous AF

37.172 (4.707, 29.562)

0.001

NS

Age > 65 (n=2,068)

Non-AF

1 (reference)

1 (reference)

NOAF

2.226 (0.961, 5.155)

0.062

2.438 (1.046, 5.680)

0.039

Previous AF

4.640 (1.860, 11.575)

0.001

3.655 (1.137, 11.753)

0.030

Ref.)

  1. Fang, M.C., et al., Gender differences in the risk of ischemic stroke and peripheral embolism in atrial fibrillation: the AnTicoagulation and Risk factors In Atrial fibrillation (ATRIA) study. Circulation 2005;112:1687-91.
  2. Dagres, N., et al., Gender-related differences in presentation, treatment, and outcome of patients with atrial fibrillation in Europe: a report from the Euro Heart Survey on Atrial Fibrillation. J Am Coll Cardiol 2007;49:572-7.
  3. Gomberg-Maitland, M., et al., Anticoagulation in women with non-valvular atrial fibrillation in the stroke prevention using an oral thrombin inhibitor (SPORTIF) trials. Eur Heart J 2006;27:1947-53.
  4. Yong, C.M., et al., Sex Differences in Oral Anticoagulation and Outcomes of Stroke and Intracranial Bleeding in Newly Diagnosed Atrial Fibrillation. J Am Heart Assoc 2020;9: e015689.
  5. Vaccarino, V., et al., Sex and racial differences in the management of acute myocardial infarction, 1994 through 2002. N Engl J Med 2005;353:671-82.
  6. Vaccarino, V., et al., Sex-based differences in early mortality after myocardial infarction. National Registry of Myocardial Infarction 2 Participants. N Engl J Med 1999;341:217-25.

2) How long is hospital stay? Can NOAF be diagnosed even if it has been a long time since the onset of ACS as long as the patient is in the hospital? The longer the time between the onset of ACS and the diagnosis of NOAF, the greater the likelihood that transient AF will be mixed with asymptomatic AF.

Response: That’s a very good point. As described in Discussion section, there was a possibility of misdiagnosis undetected silent AF as NOAF in our study. We fully agree with your opinion that patients with a longer hospital stay are more likely to be diagnosed with silent AF. In this study, NOAF patients tended to have shorter and longer hospital stays compared to non-AF and previous AF patients, respectively; however, there were no significant differences in mean hospital stays among three groups (non-AF [45.5 days] vs. NOAF [36.8 days] vs. Previous AF [26.9 days], p=0.708). Moreover, the hospital stays between men and women did not significantly differ (men [45.1 days] vs. women [44.5 days], p=0.928). The longer hospital stay of NOAF patients is speculated to be associated with more complicated hemodynamic condition of NOAF compared to previous AF (1)

Ref.)

  1. Lau, D.H., et al., New-onset atrial fibrillation and acute coronary syndrome. Expert Rev Cardiovasc Ther 2010; 8:941-8.

Minor comments

1) When were the echo data and blood test data obtained? Please describe it in Method section or in Table 1.

Response: That’s a very important point.

Laboratory data was obtained at admission and echocardiographic data was obtained within 24 hours of admission. This sentence was inserted in Method section.

2) Is there a measure of infarct size such as CK?

Response: That’s a very good point.

As mentioned above, all laboratory data including CK, CK-MB and Troponin I that has been used to diagnose AMI was obtained at admission. These biomarkers also have shown a good correlation with the extent of myocardial injury (infarct size) known as important prognostic factor in patients with AMI (1). In our study, CK, CK-MB and Troponin I were regularly checked during hospitalization in all patients presenting with AMI and peak levels of these biomarkers were obtained. As shown in revised Table 1, the variables of CK, CK-MB and Troponin I were added and we found that the peak levels of CK, CK-MB and Troponin I were significantly higher in women compared to men.

Rec)

  1. Chia S, et al. Utility of Cardiac Biomarkers in Predicting Infarct Size, Left Ventricular Function, and Clinical Outcome After Primary Percutaneous Coronary Intervention for ST-Segment Elevation Myocardial Infarction. J Am Coll Cardiol Intv 2008;1:415-23.

3) Why do so few patients with AF take anticoagulation?

Response: That’s a good point. As mentioned in Discussion section, our data were collected between 2004 and 2014, and current risk stratification concept using the CHA2DS2-VASc score for AF was not applied at that time. Although the CHADS2 score had been used to guide for anticoagulation for AF patients since 2001 (1), this score tended to underestimate the stroke risk compared with CHA2DS2-VASc score (2). Moreover, oral anticoagulants for Asian population with AF were less likely to be prescribed due to the high risk of hemorrhage (3). Considering that our study patients underwent PCI for AMI and took dual antiplatelet agents, it is speculated that there would be concerns about bleeding risk in prescription of triple therapy, including dual antiplatelet agents and oral anticoagulants.

Ref.)

  1. Gage BF, et al. Validation of clinical classification schemes for predicting stroke: results from the National Registry of Atrial Fibrillation. JAMA 2001l285:2864-70.
  2. Chao TF, et al. CHADS2 and CHA2DS2-VASc scores in the prediction of clinical outcomes in patients with atrial fibrillation after catheter ablation. J Am Coll Cardiol 2011;58:2380-85.
  3. Sabir I, et al. Oral anticoagulants for Asian patients with atrial fibrillation. Nature Review Cardiology. 2014;11:290-303.

4) Is P<0.0001, P<0.001? Only this part has one more digit.

Response: Thank you for your kind consideration.

As you pointed out, P<0.0001 was changed to P<0.001.

Round 2

Reviewer 2 Report

The authors have comprehensively amended their manuscript in line with the reviewers findings. As a result, I feel the manuscript is much improved. I therefore wish to congratulate them on the paper and on their considerable effort on the revision.